# Entropy as a Measure of Consistency in Software Architecture

**DOI:** 10.3390/e25020328

**Published:** 2023-02-10

**Authors:** Stanislaw Jerzy Niepostyn, Wiktor Bohdan Daszczuk

**Affiliations:** 1School of Computer Science & Technologies, University of Economics and Human Sciences in Warsaw, 01-043 Warsaw, Poland; 2Institute of Computer Science, Warsaw University of Technology, 00-665 Warsaw, Poland

**Keywords:** software architecture, consistency rules, IT system design, entropy, consistency assessment

## Abstract

In building software architectures, the relations between elements in different diagrams are often overlooked. The first stage of building IT systems is the use of ontology terminology, not software terminology, in the requirements engineering process. Then, when constructing software architecture, IT architects more or less consciously however introduce elements that represent the same classifier on different diagrams with similar names. These connections are called consistency rules and are usually not attached in any way in a modeling tool, and only a significant number of them in the models increase the quality of the software architecture. It is mathematically proved that the application of consistency rules increases the information content of software architecture. Authors show that increasing readability and ordering of software architecture by means of consistency rules have their mathematical rationale. In this article, we found proof of decreasing Shannon entropy while applying consistency rules in the construction of software architecture of IT systems. Therefore, it has been shown that marking selected elements in different diagrams with these same names is, therefore, an implicit way to increase the information content of software architecture while simultaneously improving its orderliness and readability. Moreover, this increase in the quality of the software architecture can be measured by entropy, which allows for checking whether the number of consistency rules is sufficient to compare different architectures, even of different sizes, thanks to entropy normalization, and checking during the development of the software architecture, what is the improvement in its orderliness and readability.

## 1. Introduction

Software architecture is a very important plan for building an IT system, in particular for such systems as safety-critical and mission-critical systems. However, it is usually believed that any gaps in the software architecture negatively affect the overall success of the entire software development project. Moreover, there is no agreed definition of software architecture to date. The structure of an IT project usually includes UML diagrams grouped into models, which are usually presented in the form of software architecture views.

It is worth mentioning that before designing the software architecture, there should always be a stage of eliciting requirements from the customer along with developing a model of the domain included in the ontology. Typically, this step transforms the requirements and their constraints into a problem ontology and tries to discover contradictions using ontology reasoning. The contradictions that have been found indicate ontological inconsistencies. These ontological inconsistencies are resolved using logical rules, while software architecture inconsistencies are resolved, for example, using database construction rules in terms of the domain model. Thus, the formal semantic notion is unimportant for software architecture but crucial for ontologies. In other words, solving the issues related to software architecture inconsistencies that we describe in the article is usually considered only after requirements are agreed upon using ontology technologies. Therefore, in the article, we omit the consideration of ontological inconsistencies, as they are beyond the scope of the problem we are considering.

In other words, in our article, we described inconsistencies in the software architecture, not inconsistencies in terms of the problem ontology coming from the domain model that represents the requirements and the domain ontology that represents the domain knowledge in a given IT project.

It has been found, since the dawn of software architecture, that the most critical gaps are inconsistencies between diagrams from different views in complex IT projects. This is the effect of grouping diagrams into views that are developed by various IT teams. Thus, the definitions used in popular architecture description standards mostly specify that “the consequence of using multiple views is the need to express and maintain consistency between these views”—ISO/IEC 42010 [1]. Until today software architects have developed a lot of standards such as the 4 + 1 architectural view model [2] constituting the base concept in the Rational Unified Process [3], ISO/IEC 42010, and Model Driven Architecture [4] or TOGAF [5], which all have some common properties: diagrams are part of specific models (e.g., data model, domain model) while models are grouped into independent views. The definitions of consistency are missing there as these standards do not directly specify dependencies between diagrams or views. Thus, presently analysts and architects mainly grapple with inconsistent architectures that restrict the progress of software development and consume excessive resources, the use of which is often critical in the project often. Currently, this is solved by accepting the existence of inconsistencies without trying to manage them, by using only part of the architecture, or even abandoning its use, especially in agile methodologies, which apparently eliminates inconsistencies in only part of the IT project by moving the problem to another part, often generating additional costs.

In this article, the authors propose yet another inconsistency management solution. The use of consistency rules in building subsequent diagrams of software architecture solves the problem of inconsistency because instead of identifying inconsistencies, subsequent elements are constructed in the next diagrams that are consistent with the previously proposed ones. Moreover, the use of such implicit consistency rules increases the information content of software architecture while simultaneously improving its orderliness and readability, the proof of which is the main contribution of the authors in this article. It is also worth noting that this increase in the quality of the software architecture can be measured by entropy, which allows us to check whether the number of consistency rules is sufficient to compare different architectures, even of different sizes, thanks to entropy normalization, and check, during the development of the software architecture, what is the improvement in its orderliness and readability.

Software architecture models are based mostly on the UML language [6]. The semantics of UML models is defined in a natural language; however, there are many attempts to formalize its semantics. UML cannot provide a straightforward way of representing a connector (association), and there is no specific construct for representing architectural styles [7]. In UML, one cannot fully define the relationships between diagrams; therefore, completeness and consistency of models composed of diagrams must be either ensured manually [8] or expressed, e.g., in OCL [9].

Therefore, when building software architectures, one often neglects the rather burdensome activity of connecting elements from different diagrams with relationships, and instead, common names for these different elements are repeatedly used to indicate some relationship between them. Thus, when constructing software architecture, IT architects more or less consciously introduce elements that refer to the same classifier on different diagrams with elements with the same names. These connections of different elements with the same names from two different diagrams are called consistency rules [10,11,12,13,14] and are usually not attached in any way to a modeling tool, and only a significant number of them in the models increase the quality of the software architecture. However, these implicit consistency rules significantly facilitate the reading of such software architecture by experienced IT architects. As a consequence, such consistency rules, often unnoticed by many IT professionals, lead to an increase in the software architecture information content, where these rules are usually seen by a few IT professionals and, at the same time, can help avoid many errors.

Our experience shows that the application of consistency rules increases the readability and information content of software architecture. However, how do we prove it mathematically? To measure the information content and readability of any diagram, or diagram layout, enriched with consistency rules, the formula for Shannon entropy, which is one of the most important metrics in information theory, can be used. Entropy measures the expected amount of information conveyed by identifying the outcome of a random trial [15]. The entropy of a more complex system will be greater than that of a less complex system. Thus, if the entropy of a diagram system without consistency rules turns out to be lower than the entropy of the same system, but enriched with consistency rules, then we will show that the application of consistency rules in a software architecture increases its information content, but also orderliness this architecture by reducing uncertainty.

The use of consistency rules increases the orderliness of a software architecture design, and the measure of this orderliness is the entropy associated with the occurrence of the same names in a given software architecture design. In this article, proof of decreasing entropy of the architecture model while applying consistency rules between various elements of different diagrams in the construction of software architecture of IT systems was carried out. This is the author’s main contribution. The uncertainty of software architecture construction is thus reduced when introducing consistency rules, and therefore, the orderliness and consistency of the entire software architecture increases.

Therefore, marking selected elements in different models using similar names is an implicit way to increase the information content of the software architecture and, at the same time, to improve its orderliness and readability. This explains the use of consistency rules in the construction of software architecture in order to cut its design time, as compared to designing without paying attention to consistency rules. The use of consistency rules also has a significant impact on the need to verify the created software architecture. The software architecture built in this way is immediately consistent, so it does not need to be verified in this respect. Thus, sometimes the great costs associated with testing the consistency of software architecture are omitted. With greater order and greater readability of the software architecture, the software coding process becomes shorter, much less complicated, and therefore less erogenous. It is also of great importance when testing software, as a faster and more complete understanding of the system’s operation, has a large impact on the time of testing as well as the number of defects found.

It is worth noting that in our article, we exaggerated the issue of showing consistency rules, which, as we showed earlier, are combinations of elements with similar names. In real projects, these consistency rules are actually hidden from the “untrained eye”. For example, an activity named “2. Register a proposal” could be the same for “a trained eye” as a use case named “UC3.1. Save proposal to registry”. To the untrained eye, such a consistency rule may go unnoticed. Moreover, there is currently no software architecture modeling tool that can automatically detect that these elements could be connected by the consistency rule and therefore be the same element but observed from two different perspectives. Thus, the discovery of the rules of consistency, which perhaps has something to do with the discovery of knowledge, has great potential but seems to be all too often overlooked due to the typical objection as being obvious. Therefore, in order not to compete with the assessment of the similarity of element names, we will assume in our article that these names must be identical if they are to be identified in the consistency rules.

The term “entropy” is used in thermodynamics, probability theory, information theory, or the theory of dynamic systems. However, as Thims [16] pointed out, Shannon’s equation regarding information theory has no relation to similar equations used in thermodynamics or statistical mechanics. In the further part of the work, the term “entropy” will be understood in the meaning assigned to it by Shannon in accordance with the formula: (1)$$\mathrm{Entropy}=-\overset{n_{1}}{\underset{i=1}{\sum }}(P_{i}\log_{2}P_{i})$$
where *n*_1_ is the total number of all elements, and *P_i_* are the probabilities of occurrence of a set *i* of possible elements.

The expressions in the Formula (1) and the following equations in this article come from the well-known equations called “Halstead complexity measures” [17]. These expressions are used in articles dealing with software metrics and even software architecture. The basic metrics of the software are: *n*_1_ is the number of distinct operators; *n*_2_ is the number of separate operands; *N*_1_ is the total number of occurrences of each operator; *N*_2_ is the total number of occurrences of the operands. Formula (1), on the other hand, concerns the entropy associated with the occurrence of operators *n*_1_ in the scope of the number *N*_1_, which is the total number of all operators. In our article, we used the formulas published by Harrison [18] and Bansiya [19]. Such identical formulas were used by Harrison, and he defined his AICC metric using the variable η_1_, instead of the number of operators *n*_1_, to denote not the number of operators but “the number of occurrences of the i-th operator in the source text”. In contrast, Bansiya in its formula also used Formula (1) in its original form, but in its formula, it used the variables *n*_1_ and *N*_1_ referring to the operators of the same name.

The formulas in this paper thus retain the convention of Halstead’s software metrics and also take into account the original form of the Harrison and Bansiya formulas. Since our article is about Computer Science, and in particular software architecture, in which we use the formulas of Halstead, Harrison, and Bansiya, it seems good practice to continue the notations described in this area of research.

Entropy may be perceived as a measure of uncertainty associated with discrete distribution with appropriate probabilities. In research on UML diagrams, it is assumed that the probability distribution of a given UML element is the quotient of the number of occurrences of a given UML element to the number of occurrences of all UML elements in a UML diagram. For example, the probability distribution for one class and for one attribute is identical and amounts to 0.5 for a UML diagram consisting of only one class (one occurrence) and one attribute (one occurrence).

For a UML diagram composed of only one class, entropy would be equal to 0 because the probability distribution for this one class would be equal to 1.0 (certain event). Hence, a UML diagram with only one class does not contain any information about the structure of a given system. Only introducing another class and linking both of them with a relation or describing this one class with attributes would bring some information to the IT architect. This is the nature of entropy, as it shows a variety of diagram elements. Entropy also indicates the complexity of such a diagram. The more diverse the elements, the greater the entropy, and thus also the information content of a given diagram. Thus, a diagram consisting of only one type of element does not present any information.

The concept of entropy has been extensively used as a measure of information content. Entropy is also used in areas such as granulation monotonicity in information systems [20], map evaluation [21], information encoding [22], sustainability transportation system structure [23], and decision systems [24].

Using the software complexity metrics with entropy, it can be proved that for the same diagram’s configuration, the decrease in entropy occurs due to the increase in the number of consistency rules (links) between the elements of these diagrams. Relationships between individual elements of diagrams should be interpreted as displaying the existence of consistency rules between elements. These rules do not introduce additional elements to the diagram configuration (relationships between elements of diagrams) but are expressed in practice through similar names of the elements being linked.

The paper is organized as follows: in Section 2, the related works on consistency rules and their classifications are summarized. Section 3 describes the software complexity metrics AICC and CDE, which are key equations in the proof. The use of the equations described above in UML diagrams is described in Section 4 and Section 5, the main proof is performed, which is the contribution of the paper. Section 6 describes an experiment performed in an industrial IT project, and Section 7 concludes the paper. Extensive excerpts from the consistency rules from another article are included in Appendix A.

## 2. Related Work

### 2.1. Consistency

To assert that something is consistent, we have to declare what it is consistent with. Software models describe the system from different points of view, at different levels of abstraction and granularity, in different notations. They may represent the viewpoints and goals of different stakeholders. However, this leads to problems in identifying and handling inconsistencies between such perspectives. Inconsistencies reveal design problems. Obviously, the earlier the problems are detected in the system design lifecycle, the lower the cost of fixing them. The research on consistency models was started by Finkelstein [25]. Finkelstein stated that inconsistency is not necessarily a bad thing, and this problem is not necessarily performed by eradicating inconsistencies but rather by supplying logical rules specifying how we should act on them. Then, in 2001, Zismann and Spanoudakis [26] described inconsistency as a state in which two or more overlapping elements of different software models make assertions about the aspects of the system they describe which are not jointly satisfiable.

A more specific definition of inconsistency was given in 2000 by Nuseibeh et al. [14] as: “any situation in which a set of descriptions does not obey some relationship that should hold between them. The relationship between descriptions can be expressed as a consistency rule against which the descriptions can be checked”.

The definition of inconsistencies in the models was provided by Spanoudakis and Zisman, indicating that the ultimate goal of management consistency is to maintain consistency during the design of the system. However, often researchers, including Straeten [27], indicate that this is not realistic in a real project, where several architects work in parallel.

Another definition of inconsistency proposed by Jurack et al. [28] also concerned the correctness of constructing the UML activity diagram. Jurack believed that the consistency of an activity diagram was preserved when all rule sequences in such a diagram were applicable. The applicability conditions for such rule sequences were given previously by Lambers et al. [29].

Finally, an original definition of diagram layout consistency was proposed in [13]. In the example of the so-called conjugated graphs, they defined (in formal logic) the consistency of the diagram’s configuration as a set of two conditions: the names of corresponding elements in two different graphs must match; the relationship and direction between elements of a given diagram must be reflected between corresponding elements in another diagram. In other words, Fryz and Kotulski found that the diagram’s configuration is consistent as long as the transformations between the diagrams take into account the connections and direction of the elements that have been transformed between them.

In the area of research on inconsistencies, an important place is also taken by the proposition of Berardi et al. [30], which is usually understood as the consistency of UML classes, in which UML classes are considered consistent only if it is possible to create an instance on their basis with all constraints, or multiples designed for these classes.

It is worth noting that despite many proposals for the definition and classification of inconsistencies in the last dozen or so years, it turns out that the definition of Spanoudakis and Zisman has become the best-known and most often quoted. Moreover, this definition, as one of the few, has been described in formal language. Most of the other definitions of inconsistency were described in informal language.

Among these many proposals for the definition and classification of inconsistencies, it is worth noting that in 2002–2003 two conferences were held in Poland on the issues of consistency of models using the UML standard [31,32]. At these conferences, it was proposed to divide the inconsistencies that violate the consistency of UML models into such types as inter-model consistency or intra-model consistency. Inter-model consistency is defined as meeting the appropriate constraints for the model but also maintaining the appropriate rules of compliance with the modeling language used. The definition of inconsistency was most often described in informal language, supported by meta-models, while the inconsistency detection algorithms were proposed in pseudo-code.

The next types of inconsistencies were proposed by Engels et al. [33]. He defined evolution consistency. Such inconsistency occurred between different versions of the same model. Regardless of these types of consistency, Engels also defined syntactic consistency and semantic consistency. The semantic consistency of the model consists of the compliance with the semantic rules described by a given UML meta-model, while the syntactic consistency consists of the compliance of the diagram with the rules of creating diagrams described by the UML standard. Horizontal consistency of software models has been defined as the possibility of implementing such models into an executable form. On the other hand, vertical consistency has been described as the consistency between a given model and its transformation at the next level of abstraction, with the last level being the implemented piece of software or the entire software. It is worth noting the vertical consistency here, which could allow the construction of architecture from abstract to implementable models.

In 2005, Mens [34] proposed to divide inconsistencies into three dimensions based on research into class, sequence, and state diagrams. The first dimension includes inconsistencies related to horizontal, vertical, and evolutionary consistency. Vertical consistency is an agreement between diagrams at different levels of abstraction, horizontal consistency is an agreement of diagrams on the same level of abstraction, and evolutionary consistency is an agreement between different versions of the same diagram. The second dimension of inconsistency was defined as syntactic consistency and semantic consistency. Semantic consistency is in compliance with the semantic rules (in natural language) defined by the UML standard, and syntactic consistency is in compliance with the specification of its meta-model. The third and last dimension of inconsistency describes the inconsistencies resulting from the inheritance hierarchy defined in object-oriented languages and consists of observational consistency (a subclass object should always behave like a superclass object) and invocation consistency (a subclass object can be used wherever it is required object of this class).

Subsequent proposals for the classification or interpretation of inconsistencies did not have a significant impact on the further development of research on inconsistencies. In the area of research on inconsistency, the focus of further development of inconsistency was the marking, specification, and application of the rules of consistency in UML diagrams.

### 2.2. Consistency Rules in UML Diagrams

UML diagram consistency rules do not have a clear and unambiguous definition so far. Most often, the definitions of consistency rules are closely related to the definition of consistency. The accepted conception of consistency rules is the relationship between elements of various UML diagrams that meet a specific consistency definition and, as a result, are considered consistent. In general, consistency rules are transformations (mappings) between elements of different models. Because relationships between diagrams require additional diagrams regardless of the modeling language, separate UML diagrams must be created to visualize consistency rules.

One of the first implementations of the rules of consistency between diagrams was proposed in 2000 by Egyed [35]. These rules were called constraints or transformations by Egyed, while he used the term “consistency rules” to denote a certain type of constraint between elements of different diagrams. However, for the purposes of this dissertation, many of the links proposed by Egyed have been interpreted as consistency rules. Egyed’s rules applied to class, state, object, and sequence diagrams but did not determine specific transformations leading to the generation of executable code or the construction of diagrams from abstract to implementable. Other consistency rules, but between sequence diagrams and state machines, were developed in 2006 by Shuzhen et al. [36]. Ha et al. [37], however, proposed the rules of consistency between nine UML diagrams without indicating specific elements of these diagrams. These suggestions were only used to improve the quality of the constructed models. Hausmann’s proposal [38] was to use consistency rules to verify the consistency of activity diagrams visualizing use cases. Sapna et al. [39] proposed the implementation of consistency rules between use case, activity, sequence, class, and state machine diagrams in the form of SQL expressions. The rules of consistency postulated by Chanda et al. [40] were to be implemented through context-free grammar using Lex and YACC programs. These rules are applied to the diagram of use cases, classes, and activities. Only Ibrahim et al. [41], Shinkawa [42], and Kang et al. [43] proposed consistency rules that could be used to generate consistent UML diagrams. Ibrahim et al. [44] proposed rules of consistency between the use case diagram and the activity diagram and then between the sequence diagram. Kang et al. indicated the rules of consistency between the sequence diagram and the activity diagram. Shinkawa, on the other hand, assumed that consistency between specific diagrams was too complex to study or manage, so he proposed a method to generate consistent UML models from the use case model. It was proposed, on the basis of the “use-case driven” method, to create an activity model from the description of use cases. In turn, objects are extracted from this model by formulating an object diagram and then a state machine diagram consistent with the previously created class diagram. Moreover, a sequence diagram is obtained from the activity model through appropriate transformations. In order to obtain consistent diagrams, Shinkawa decided to map the scenarios of individual use cases to the corresponding models in CPN (Colored Petri Net) notation, and then from such a model, a CPN activity model is created, on the basis of which activity diagrams, classes, state machines and sequence. In the above method, the main assumption was that the UML diagrams obtained from the CPN activity model are consistent because the CPN activity model is also consistent.

At the end of this summary, it is worth mentioning that many studies on inconsistencies, along with the area of application of consistency rules [34,36,39], have come to interesting proposals for defining and applying individual properties of various UML models in the field of software development. In addition, recent publications in this area, Torre et al. [10,45], seem to aim to develop a full list of UML diagram consistency rules according to the new proposals for their classification after Allaki et al. [46]. Thus, in 2016 Torre et al. [10] showed 116 UML consistency rules gathered from different authors, and in 2019 Niepostyn [12] published 87 original UML consistency rules.

After these publications, it seems that the next step should be methods and algorithms enabling the easy and clear implementation of consistency rules in software architecture.

It is worth adding that many authors try to describe the consistency of software architecture, but to the best of our knowledge, no one has tried to measure and thus compare the software architecture of IT systems.

## 3. AICC and CDE Software Complexity Metrics

Before we move on to describe the proof of the decrease in entropy when applying consistency rules, we would like to present the entropy measures that will be used later in the article in our proof.

One of the first metrics based on estimating the information content (entropy) of the software data structure was the AICC [18] (Average Information Content Classification) and the CDE [19] (Class Design Entropy) metrics, which calculated the complexity of the software code, the first referring to structural languages (in particular PL/I) and the other object-oriented languages. It is worth noting that the AICC and CDE metrics have been proposed for estimating the source codes (measuring complexity). The proposed mathematical formulas of these metrics were used in their later applications to assess, among others, UML diagrams.

The Average Information Content Classification metrics (AICC) described the complexity of the software based on the concept of entropy and was proposed to estimate the complexity of the source code. The AICC metrics include the following parameters: *N*_1_ is the total number of (non-unique) symbols of the language used in the code, and $f_{i}$, where $1\leq i\leq \eta_{1}$, is the number of occurrences of the *i*-th language symbol appearing in the source code. The formula for calculating the AICC metrics is given below.


(2)
$$\mathrm{AICC}=-\sum_{i=1}^{\eta_{1}}\frac{f_{i}}{N_{1}}\log_{2}\frac{f_{i}}{N_{1}}$$


The interpretation of this metric is that a program with a higher value of a metric should be less complex (complex) than a program with a lower value of this metric. The AICC values are not additive, nor is the comparison of its values for two different modules not meaningful, but already these values for the same module could indicate the justification for using, for example, some complex software libraries. For large systems, the AICC metric can take values between 1.7 and 5.1. In AICC-based diagrams describing UML diagrams, the elements from which a given diagram can be built are used as language symbols, whereas $\eta_{1}$ means the number of groups of elements of the same type.

The Class Design Entropy metrics, such as the AICC metrics, have also been proposed for estimating the source code. An identical formula was proposed for calculating the value of this metric as for the AICC metric, where instead of all language symbols, only the names of identifiers appearing in the tested part of the software (class definitions) such as class name, variable, constant, class, and symbols characteristic for a given language are included. On the other hand, symbols characteristic of a given language, such as keywords or operators, are omitted. The proposed CDE metrics include the following parameters: *N*_1_ is the total number of occurrences of identifiers (non-unique) used in the definition of the tested class, and ${\hat{f}}_{i}$, where $1\leq i\leq n_{1}$ is the number of occurrences of the *i*-th identifier in the definition of this class, $n_{1}$ is the number of group identifiers with the same name (unique identifier names) of the class definition being examined. The formula for calculating the CDE metrics, which is identical to (2), is given below, but the symbols used have different meanings.
(3)$$\mathrm{CDE}=-\sum_{i=1}^{n_{1}}\frac{{\hat{f}}_{i}}{N_{1}}\;\log_{2}\frac{{\hat{f}}_{i}}{N_{1}}$$

In the diagrams based on the above formula, describing UML diagrams, the name of the UML element of the analyzed diagram is usually taken as software code identifiers, whereas $n_{1}$ means the number of groups of UML elements having similar names.

The AICC and CDE metrics have been used in the present article to demonstrate that the application of consistency rules, understood as linking elements of identical interpretation in any independent diagram, results in a decrease in its entropy, i.e., an increase in the information content of the model and an increase in its orderliness.

## 4. Preliminaries

In this section, we will show the application of the above-described formulas to UML diagrams. Figure 1 shows two configurations with two UML diagrams. On the left is a configuration with independent diagrams, and on the right is a configuration with related diagrams. The configuration shown on the left has four occurrences of independent UML elements (two occurrences of *UML Activity* elements named “a” and “c”, one occurrence of *UML UseCase* element named “d” and one occurrence of *UML ControlFlow* element without a name—an arrow connecting UML elements “a” and “c”). Thus, according to (2), parameter *N_1_* is the number of all occurrences of UML elements (4 UML elements), parameter $\eta_{1}$ is the number of all types of elements in the diagram (3 types of elements: *UML Activity*, *UML UseCase*, *UML ControlFlow*), and the number of occurrences of individual UML elements are as follows: $f_{UML\;Activity}=2$ (2 elements of *UML Activity*); $f_{UML\;UseCase}=1$ (1 element of *UML UseCase*); $f_{UML\;ControlFlow}=1$ (1 element of *UML ControlFlow*).

On the other hand, the configuration shown on the right has two independent elements (*UML Activity* element named “a” and unnamed *UML ControlFlow* connection) and one dependent element (element named “c” appearing as *UML Activity* element in the top diagram and the same element occurring as *UML UseCase* in the bottom diagram). The connection drawn with a dashed line and named “link” is not part of the configuration but indicates the consistency rule. Thus, according to (3), parameter *N*_1_ means the number of all UML elements (4 non-unique identifiers, i.e., all UML elements), parameter $\eta_{1}$ is the number of all elements on the diagram having different names (3 unique identifiers: “a”, “c”, and an unnamed identifier). Moreover, the number of occurrences of individual UML elements with different names is as follows: ${\hat{f}}_{a}=1$ (1 UML element with the identifier “a”); ${\hat{f}}_{c}=2$ (2 UML elements with identifier “c”); ${\hat{f}}_{unnamed}=1$ (1 UML element with id unnamed).

It is worth noting that unrelated UML elements occur only once in a diagram system, whereas UML elements related in a given configuration occur many times in the form of various UML elements (e.g., the UML element named “c” occurs once as part of the *UML Activity*, and for the second time as *UML UseCase* element).

Thus, to calculate the entropy for the left unrelated system shown in Figure 1, we use the entropy formula (2) AICC = $-\sum_{i=1}^{\eta_{1}}\frac{f_{i}}{N_{1}}\log_{2}\frac{f_{i}}{N_{1}}$ − (1/4)log(1/4) {element „a”} − (1/4)log(1/4) {element unnamed} − (1/4)log(1/4) {element ”c”} = −(1/4)log(1/4){element ”d”} = −(4*1/4)log(1/4) ≈0.6, because “*N*_1_ is the total number of (non-unique) symbols of the language used in the code (there are 4 symbols), and $f_{i}$, where $1\leq i\leq \eta_{1}$, is the number of occurrences of the *i*-th language symbol appearing in the source code (each element has the value 1 for $f_{i}$)”.

Whereas for the system with consistency rules, we use the entropy formula (3) CDE =$\sum_{i=1}^{n_{1}}\frac{{\hat{f}}_{i}}{N_{1}}\log_{2}\frac{{\hat{f}}_{i}}{N_{1}}=$−(1/4)log(1/4) {element ”a”} − (1/4)log(1/4) {element unnamed} − (1/4)log(2/4) {element ”c” in UML Activity Diagram} − (1/4)log(2/4) {element ”c” in UML USE Case Diagram} = −(2*×1/4)log(1/4) − (2/4)log(2/4) = −(1/2)(log(1/4) − log(2/4)) = log2 ≈ 0.3, because “*N*_1_ is the total number of occurrences of identifiers (non-unique) used in the definition of the tested class (there are 4 occurrences of identifiers), and ${\hat{f}}_{i}{}_{}$, where $1\leq i\leq n_{1}$ is the number of occurrences of the *i*-th identifier in the definition of this class, $n_{1}$ is the number of groups identifiers with the same name (unique identifier names) of the class definition being examined (two elements have a value of 1 for ${\hat{f}}_{i}$, and two elements, which are instances of the same element, have a value 2 for ${\hat{f}}_{i})$”.

## 5. Proof of the Decrease in Entropy When Applying Consistency Rules

Below is proof of the decrease in entropy when linking UML elements of different diagrams using identical names, i.e., it will be proved that $E_{indep}>E_{dep}$, where $E_{indep}$ is the entropy of the configuration of independent diagrams, and $E_{dep}$ is the entropy of the configuration of dependent diagrams using consistency rules. The formula for the AICC metrics described in (2) will be used, as well as the formula for the CDE metrics given in (3).

Assuming that the entropy $E_{indep}$ for an unrelated configuration results from the formula for AICC and is given by the formula:(4)$$E_{\mathrm{indep}}=\mathrm{AICC}=-\sum_{i=1}^{\eta_{1}}\frac{f_{i}}{N}\log_{2}\frac{f_{i}}{N},\;N\in N^{+},f_{i}\in N^{+}.\;$$
where the symbol *N* denotes the number of all elements of the configuration of independent (unrelated) diagrams, and the symbol $\eta_{1}$ denotes the number of occurrences of elements of type *i*, and $f_{i}$ is the number of occurrences of elements of the *i*-th of the type *i* = 0, 1, 2 … for independent diagrams.

On the other hand, entropy $E_{dep}$ for the same related configuration results from the formula for CDE and is given by the following formula:(5)$$E_{\mathrm{dep}}=\mathrm{CDE}=-\sum_{i=1}^{\eta_{2}}\frac{{\hat{f}}_{i}}{N}\log_{2}\frac{{\hat{f}}_{i}}{N},\;N\in N^{+},{\hat{f}}_{i}\in N^{+}$$

The symbol *N* denotes the number of all configuration elements of related diagrams, and the symbol $\eta_{2}$ represents the number of occurrences of elements with the given name *i*, and ${\hat{f}}_{i}$ the number of occurrences of the *i*-th element with the given name *i* = 0, 1, 2 … For the configuration of independent diagrams, there is equality: $f_{1}=f_{2}=\cdots =f_{\eta_{1}}=1$. By inequalities $0\leq id\leq \eta_{2}$ let us indicate the number of elements (in the configuration of related diagrams) such that ${\hat{f}}_{k}=1$ for related diagrams. Without loss of generality, we can assume that (the dependent layout elements from 1 to id occur only once):(6)$${\hat{f}}_{1}={\hat{f}}_{2}=\cdots ={\hat{f}}_{\mathrm{id}\;}=1\;$$

In addition, equality is satisfied (the number of elements in both layouts is equal):(7)$$\sum_{i=1}^{\eta_{1}}f_{i}=\sum_{i=1}^{\eta_{2}}{\hat{f}}_{i}=N$$
and inequality $\eta_{1}>\eta_{2}$. Thus, if we substitute Formulas (4) and (5), respectively, in the main equation $E_{indep}>E_{dep}$, then:(8)$$E_{\mathrm{indep}}-E_{\mathrm{dep}}=-\sum_{i=1}^{\eta_{1}}\frac{f_{i}}{N}\log_{2}\frac{f_{i}}{N}\;+\;\sum_{i=1}^{\eta_{2}}\frac{{\hat{f}}_{i}}{N}\log_{2}\frac{{\hat{f}}_{i}}{N}\;=\;-\sum_{i=1}^{id}\frac{f_{i}}{N}\log_{2}\frac{f_{i}}{N}-\sum_{j=id+1}^{\eta_{1}}\frac{f_{j}}{N}\log_{2}\frac{f_{j}}{N}+\sum_{i=1}^{id}\frac{{\hat{f}}_{i}}{N}\log_{2}\frac{{\hat{f}}_{i}}{N}+\sum_{j=id+1}^{\eta_{2}}\frac{{\hat{f}}_{j}}{N}\log_{2}\frac{{\hat{f}}_{j}}{N}\;=\;-\sum_{i=1}^{\mathrm{id}}\frac{1}{N}\log_{2}\frac{1}{N}-\sum_{i=\mathrm{id}+1}^{\eta_{1}}\frac{1}{N}\log_{2}\frac{1}{N}+\sum_{i=1}^{\mathrm{id}}\frac{1}{N}\log_{2}\frac{1}{N}+\sum_{j=\mathrm{id}+1}^{\eta_{2}}\frac{{\hat{f}}_{j}}{N}\log_{2}\frac{{\hat{f}}_{j}}{N}=\;-\sum_{i=id+1}^{\eta_{1}}\frac{1}{N}\log_{2}\frac{1}{N}+\sum_{j=id+1}^{\eta_{2}}\frac{{\hat{f}}_{j}}{N}\log_{2}\frac{{\hat{f}}_{j}}{N}$$

Equations (6) and (7) show that:(9)$$\sum_{j=\mathrm{id}+1}^{\eta_{2}}{\hat{f}}_{j}=\eta_{1}-\mathrm{id}\;$$

Then:(10)$$-\overset{\eta_{1}}{\underset{i=id+1}{\sum }}\frac{1}{N}\log_{2}\frac{1}{N}=-\frac{1}{N}\log_{2}\frac{1}{N}\overset{\eta_{1}}{\underset{i=id+1}{\sum }}1=-\frac{1}{N}\log_{2}\frac{1}{N}\left(\eta_{1}-id\right)=-\frac{1}{N}\log_{2}\frac{1}{N}\overset{\eta_{2}}{\underset{j=id+1}{\sum }}{\hat{f}}_{j}\;=-\sum_{j=id+1}^{\eta_{2}}\frac{{\hat{f}}_{j}}{N}\log_{2}\frac{1}{N}$$

Thus, Equation (8), reduced to a common upper limit, can be written
(11)$$as:\;\sum_{j=\mathrm{id}+1}^{\eta_{2}}\left(\frac{{\hat{f}}_{j}}{N}\log_{2}\frac{{\hat{f}}_{j}}{N}-\frac{{\hat{f}}_{j}}{N}\log_{2}\frac{1}{N}\right)\;\sum_{j=id+1}^{\eta_{2}}\frac{{\hat{f}}_{j}}{N}\left(\log_{2}\frac{{\hat{f}}_{j}}{N}-\log_{2}\frac{1}{N}\right)$$

Because $\forall j\in \left\{id+1,\ldots ,\eta_{2}\right\}$ then the inequality ${\hat{f}}_{j}>1$ occurs and from monotonicity of the function $\log_{2}x$ we obtain:(12)$$\sum_{j=\mathrm{id}+1}^{\eta_{2}}\frac{{\hat{f}}_{j}}{N}\left(\log_{2}\frac{{\hat{f}}_{j}}{N}-\log_{2}\frac{1}{N}\right)>0\;$$

What ends the proof that $E_{indep}>E_{dep}$.

## 6. An Example of the Application of Consistency Rules

This section introduces the same piece of design that was modeled without applying consistency rules and then shows what a software architecture design should look like if consistency rules have been applied.

Figure 2 shows a fragment of a project implemented by the Ministry of the Interior in 2014 in all state administration offices as the System of State Registers [47], which is an organizational and technical solution used to keep public registers, such as: PESEL Register (Universal Electronic System for Registration of the Population); Register of Identity Cards (Polish identity card); Civil Registry (civil status and marital status); System of State Decorations (Orders, decorations, and medals of Poland); Central Register of Objections (organ and/or tissue retrieval). The project did not apply the consistency rules, while the audit carried out by the author of this article in 2014 showed significant shortcomings of the system, which resulted in postponing its deployment for half a year in order to improve its design.

Due to the large deficiencies in the software architecture of this project, the authors decided to place all diagrams from this project in the proprietary software architecture model, called e-CMDA, divided into four views: context, business, system, and development. The original project documentation was not presented in this way, and the individual diagrams were not in any way arranged. In the e-CMDA software architecture concept, each view contains two layers with corresponding diagrams. In the absence of diagrams from the above-mentioned project, the authors inserted an inscription with the appropriate text included in the project documentation instead of the UML model or with the text “X” when there was no information about a similar artifact in the project. The presented software architecture is the authors’ personal contribution to the improvement of the original software architecture presented in the design documentation, and the placement of individual diagrams in an orderly manner indicates an additional advantage of the entropy concept in the software architecture presented by the authors.

In Figure 2, to improve the readability of the models, we have also placed comments next to selected elements of individual UML diagrams so as to better use the definition of the consistency rule, which states that it is a combination of elements with the same names from different diagrams. It is worth noting that the very fact of arranging the diagrams from the most abstract (Context Diagram) to the most technical (Component Diagram) introduces a large ordering of the software architecture, compared to the software architecture consisting of randomly shown diagrams in the design documentation, as is often the case in many IT projects.

The contents of the diagrams are too small to be readable, but we show the general design just to enumerate the diagrams that are subject to the application of consistency rules.

Comparing the number of elements from the improved diagram in Figure 2 with the number of elements from the diagram with the consistency rules in Figure 3, it turns out that the number of these elements in these figures is equal. The evidence carried out in this article shows that by taking into account the elements with the same names that they are one and the same element, and thus reducing the number of different elements on the software architecture model, we simultaneously reduce entropy, and thus increase the readability of the software architecture and its orderliness.

And in fact, by reducing the number of different elements, the software architecture becomes more readable, and the ordering of these elements increases because it is known which elements are based on which. Thus, by applying the consistency rules, we simultaneously reduce the entropy of the software architecture, which entails the fact that the software architecture becomes more readable and understandable, and we see order and a rational explanation of the location of individual diagrams and their elements in the entire software architecture.

To better support the examples cited, the entropy of the software architecture parts shown in Figure 2 and Figure 3 was measured for the following diagrams: context diagram, business use case diagram, process decomposition diagram, business use case realization diagram, system use case diagram, business class diagram, system use case realization diagram, internal system use case diagram, sequence diagram, and component diagram. The entropy in Figure 2 for software architecture without consistency rules applied is 3.758, while the entropy in Figure 3 for software architecture using consistency rules is −3.699—Table 1. In both figures, the selected diagrams include 176 elements. For software architecture without consistency rules, 21 element groups were identified, and for software architecture with consistency rules, 54 element groups were identified. Thus, the described example of two identical software architectures supports the proven thesis.

The software architecture shown in Figure 3 seems more obvious and rational. It can be deduced from this architecture how the next diagrams should be developed during the software implementation. It is worth adding that in many IT projects, component diagrams are created at the beginning of their implementation, which shows a more technical approach to the possibility of building the designed IT system. The software architecture model shown above shows that the component diagram should be one of the last diagrams created in an IT project because it should be created after designing business processes, operation of screens, and showing the relationships of the designed system with individual components of the entire system. It is also worth noting that the presented software architecture model indicates the most necessary UML diagrams and their order of construction. Often, in many IT projects, diagrams are created that do not contribute too much to the project development of the entire software architecture and can even cause many inconsistencies and ambiguities in other diagrams. Therefore, the proposed method of building software architecture indicates the most necessary diagrams and their arrangement, which, combined with the use of consistency rules, locates the e-CMDA method in the group of methods for fast, consistent, and complete software architecture design. The demonstrated evidence of entropy reduction indicates further benefits of using the e-CMDA method.

Table 1 shows some information about two industrial IT projects. The first of these projects, System of State Registers, was implemented for three years with rather poor software architecture, and no consistency rules were applied. The data relate only to selected fragments of the software architecture, while the entropy with consistency rules was calculated for a hypothetical case in which the consistency rules would be applied in the project as stated in the above software architecture models. The second system, PKWD SingleWindow [48], was implemented for the Ministry of Finance—National Revenue Administration, with the participation of the author as a software architecture designer using consistency rules. The architecture design was completed at the beginning of 2019, and the entire system was made available to the business on 30 April 2022. The table above shows that the application of consistency rules has significantly accelerated the software architecture design period and significantly reduced the size of the analytical team, while in the second case, the number of software architecture elements and their detail was much greater than in the first project.

## 7. Conclusions

The above evidence shows that for the configuration of diagrams with consistency rules, created by assigning identical names to diagram elements, the entropy value of the system decreases compared to the configuration with the same number of elements but without consistency rules. Thus, the introduction of consistency rules (links between elements by assigning similar names to some elements) reduces the uncertainty of information and consequently increases the orderliness and consistency of the whole model. Typically, this method of building software architecture is performed by experienced IT architects. In this article, therefore, a rationale for the practices of experienced IT architects was presented. The use of consistency rules also has a significant impact on the need to verify the created software architecture. The software architecture built in this way is immediately consistent, so it does not need to be verified in this respect. Thus, sometimes the great costs associated with testing the consistency of software architecture are omitted. With greater order and greater readability of the software architecture, the software coding process becomes shorter, much less complicated, and therefore less erogenous. It is also of great importance when testing software, as a faster and more complete understanding of the system’s operation, has a large impact on the time of testing as well as the number of defects found.

A very important conclusion implied from our proof is the fact that when developing software architecture, consistently apply the rules of consistency in such a way that in subsequent details of the project, in which there are more and more elements, the entropy should be measured each time and care should be taken to ensure that to keep this entropy down all the time. The use of consistency rules supports this direction of software architecture development, as each added consistency rule reduces the entropy of the entire software architecture. Thus, when new elements appear in the next iteration of software architecture development, measuring its entropy allows assessing whether the introduction of new elements goes with a sufficient number of introduced consistency rules. So it allows assessing whether the detailed software architecture is no less consistent than its more abstract version.

Ultimately, the following advantages can be formulated, which can be achieved by consciously applying the consistency rules in constructing software architecture:Consistency rules affect faster, optimal, and reliable development of software architecture because they are certain patterns that are easy to implementConsistency rules can very quickly verify deficiencies in the software architectureThere is no need to verify the consistency of the software architecture constructed in accordance with the consistency rulesGreater readability and more explicit rationale for building software architecture shortens the time of its interpretation and therefore reduces the time of coding and testing IT systemsConsistency rules should be included in software architecture projects (e.g., in modeling tools) to remove ambiguities.

## Figures and Tables

**Figure 1 entropy-25-00328-f001:**
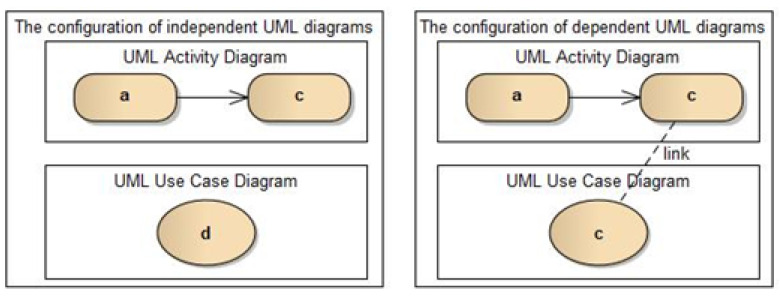
Configuration with two UML diagrams with a minimum number of element.

**Figure 2 entropy-25-00328-f002:**
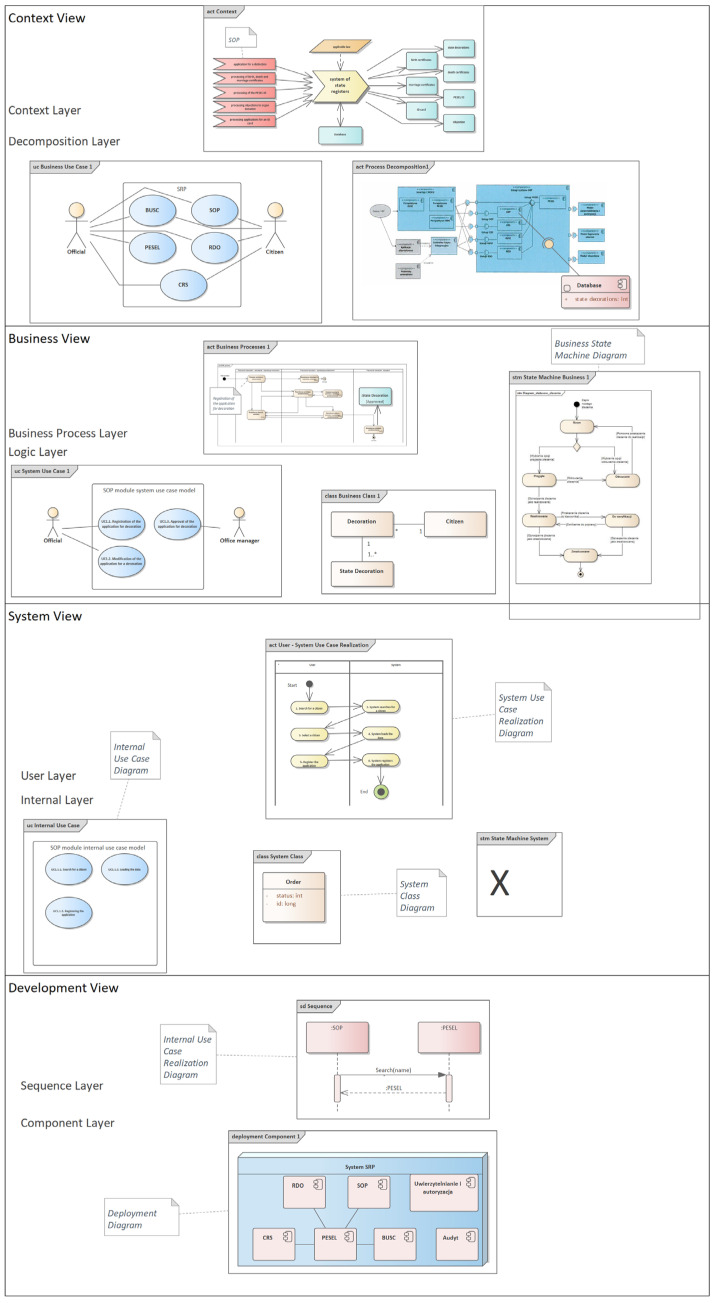
System of State Registers—UML design.

**Figure 3 entropy-25-00328-f003:**
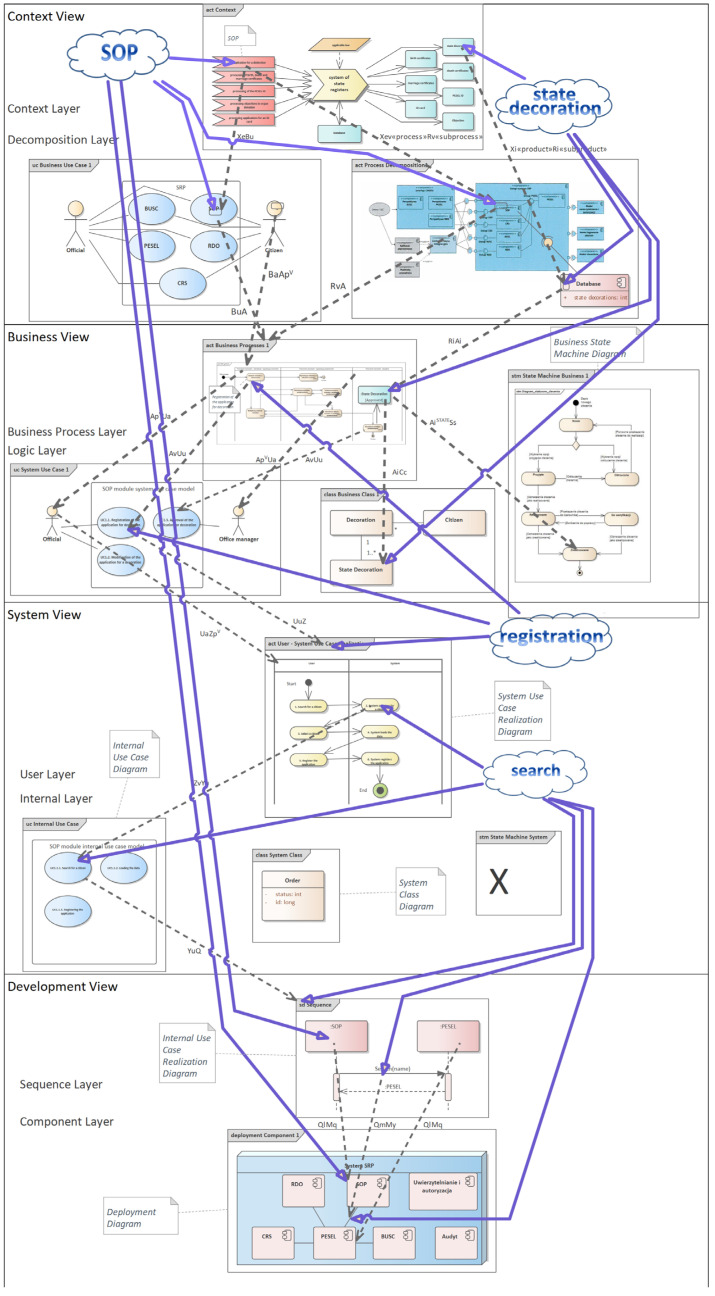
System of State Registers—consistency rules version.

**Table 1 entropy-25-00328-t001:** Information about industrial IT projects.

Name	System of State Registers	PKWD Single Window
Implementation period	2012–2015	2018–2019
Software architecture design period	2 years	0.5 year
Number of people involved in the software architecture design	8	1
Number of elements	176	5010
Number of consistency rules	0 (95)	427
Entropy without consistency rules	3.758	3.514
Entropy with consistency rules	3.699	3.092

## Data Availability

The availability of this data is not restricted. The data was obtained on the basis of the authors’ models developed for the needs of the audit of the IT system commissioned by the Polish government, co-financed by the European Union from the European Regional Development Fund (ERDF) under the Operational Program: Innovative Economy 2007–2013 Priority Axis 7-Information Society-development of electronic administration. The report, after its adoption, became public information: https://www.gov.pl/web/cyfrizator/materialy-dokumentujace-sposob-wykonowania-przed-coi-umowy-z-ministrem-spraw-wewnetrznych-dot.- construction-of-the-state-register-system-srp-#:~:text=Report%20of%20execution%20contract%20no.%20674/DEP/4.8/2014 (accessed on 6 February 2023).

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
