# Peer review of "Entropy as a Measure of Consistency in Software Architecture"

_entropy, 2023, doi:10.3390/e25020328_

Round 1

Reviewer 1 Report (New Reviewer)

The reviewed work appears to be an interesting approach to applying entropy to a practical problem. However, I believe that it contains a lot of shortcomings. I list some examples below.

Major remarks:

I. The article does not include clear goal (it should be presented in the abstract or in the introduction). We can guess, but it should be clarified and easy to understand. 

* What is the aim of the paper, and what is the contribution of the paper? What will be achieved? It should be written in the first part or in the abstract.

II. The introduction should also include information about the sections that are in the article and what they contain.

III. The work has no experiments -- it is theoretical, but in such a situation it should be as elaborate as possible in this respect, and it is not.

* Why is the total number of all elements in equation (1) called n_1 and not, for example, simply n? Similarly, N_1 and ?_1 in equation (2) (note ? --not to be confused with n-- is not defined, the definition only appears in equation (4)), whereas in equation (3) n_1 is explained differently (although I know it means the same thing, the description is different).

* The calculations in section 4 are not readable, e.g.

CDE = -(2*1/4)log(1/4)-(2/4)log(2/4) = -(1/2)(log(1/4)-log(2/4))=log2=~ 0.3

It should be written, as equations (in TeX "content $equation$") content allows you to write the equation, as text content.

Author Response

Please find the responses attached

Reviewer 2 Report (New Reviewer)

Several minor typos  

Recommend that the authors refer to literature that discusses ontologies and how they are needed and used. Models built without an ontology will suffer the chaos mentioned. However, most systems engineering efforts begin by defining their ontology.

The introduction discusses several issues related to ontological inconsistencies and explains that these problems are common. Please provide data/references that support these statements because they do not represent well-planned and executed systems engineering as commonly performed today.

Many software modeling languages and the tools that support them create views from the components and relationships within the model. Consequently, inconsistencies in naming conventions and relationships result from model problems and not from views of the model. Statements in line 52-54 is standard practice when systems are properly developed.

Line 45 & 46 mentions accepting inconsistencies rather than managing them. As noted above, the reason for creating ontologies is precisely to prevent and manage inconsistencies. 

Lines 58-64, mostly true for UML but SysML does offer additional capability and software as well as other systems have moved increasingly to SysML. A UML/SysML "association" is a weak-semantic relationship. Other relationships such as composition, aggregation, decomposition, generalization/specialization, object and control flows, etc. are more often used because they have more refined semantics. Diagram relationships are also possible if ontological metamodels are used. 

Line 65: Most UML/SysML tools support the reuse of relationships. That is, a particular semantic connection between two components in one diagram are automatically repeated in another diagram. 

Line 72 - see above comment. This is not true in modern tools.

Line 381: is the intent that unrelated elements have only one set of relationships with other elements? Wouldn't it be possible to show an unrelated element multiple times in different views in which those relationships do not change?

Please provide explain the difference between a consistency rule and enforcement of a metamodel/ontology. If these are the same then it would help readers to make that point. If they are different then please explain the difference and what the deficiencies are in metamodels and ontologies.

Author Response

Please find the responses attached

Round 2

Reviewer 1 Report (New Reviewer)

The paper is improved essentially. I agree with changes made according to my remarks.

This manuscript is a resubmission of an earlier submission. The following is a list of the peer review reports and author responses from that submission.

Round 1

Reviewer 1 Report

The paper proposes and discusses a proof of decreasing entropy of the architecture model while applying consistency rules between various elements of different diagrams in the construction of software architecture of IT systems was carried out. Authors claim that the uncertainty of software architecture construction is thus reduced when introducing consistency rules and therefore the orderliness and consistency of the entire software architecture increases.   Although the work seems interesting and may have potential, it is currently very preliminary to being published as a journal article. Accordingly, the authors should provide for an adequate extension to the work: complete discussion of the state of the art, outline of the contribution, as well as advancement with respect to the state of the art, and possible discussion of a real case study to highlight the application benefits of using consistency rules.

Author Response

We would like to thank the reviewer for the thorough reading of our article. We would like to point out however that the main contribution of our article is to provide the proof, not to describe a case study, as the evidence covers a wide range, if not all, of models involved in software architecture design. This proof providing seems to be an act convincing to apply consistency rules in designing software architecture, although we noted at the beginning of our article that these consistency rules are widely used by IT architects who apply them more or less consciously. In the next version of our article, however we introduced an additional chapter related to the extension of the description of this area.

We deemed that a case study in a full-evidence article is not usually required in the Entropy journal as in the referenced articles "A Proof of Komlós Theorem for Super-Reflexive" or "A New Proof of the Existence of Nonzero Weak Solutions of Impulsive Fractional Boundary Value Problems" released in 2020. Such details are available and could be easily and quickly attached to the paper, so required by the Editor.

Reviewer 2 Report

The paper postulates that a UML architecture description that uses the same names for certain aspects in different places has a lower entropy than a UML architecture description that uses different names for the same concepts. Entropy is equated with complexity. The paper confirms this assumption by means of a mathematical proof based on the entropy formula.

The proof relies exclusively on the idea that when fewer different identifier names are used in an architecture description, the entropy is less. Sometimes, even immediately obvious things might need a mathematical proof, I wonder if this is the case here. Please explain why a proof is needed. Isn’t this just too obvious?

Please further explain how strict consistency rules are applied. Does applying a consistency rule mean that certain “duplicated” identifiers are completely omitted or are just some occurrences of them omitted? In the first case, the entropy reduction is intuitively trivial because a term is completely removed. In the second case, what happens if a and b are identifiers of a linked aspect. Let us assume that a is more frequent than b (i.e., fa > fb). If a single use of a is replaced by a single use of b (, the entropy will, in fact, increase. How does this relate to your proof? For example, if a is used 3 times and b is used once, entropy is lower than if a and b are used 2 times each.

Introduction: I recommend to clearly point out contributions and add an overview of the paper. Possibly move discussion of related work to a related work section.

Lines 46-52: Please provide more a concrete example to improve understandability of the context of your discussion.

Line 53-56: “For a UML diagram composed of only one class, entropy would be equal to 0, because the probability distribution for this one class would be equal to 1.0 (certain event). Hence, a UML diagram with only one class does not contain any information about the given system.”

This does not make sense. A diagram with only one class has entropy = 0, but that does not mean it has information = 0. In my understanding, entropy is the amount of information you must put in to fully understand. A diagram with zero entropy is immediately fully understandable, i.e., it does contain all necessary information.

Line 58: “Using the software complexity metrics with entropy, it can be proved that for the same diagrams configuration, the decrease in entropy occurs due to the increase in the 58number of consistency rules (links) between the elements of these diagrams.

Unclear what this means. “diagrams configuration” = “diagram’s configuration”?

The singular of metrics is metric (e.g., Line 84: “The AICC metrics” -> metric). Further occurrences across the paper, e.g.: Line 75: “One of the first metrics based on estimating the information content (entropy) of the software data structure was the” -> “Among the first … were…”?  Line 96: has -> have.

Line 97: What do you mean by “estimating the source code”?

Line 125 and following: “The configuration shown on the left has 4 occurrences of independent UML elements (two occurrences of UML Activity elements named "a" and "c", one occurrence of UML UseCase element named "d" and one occurrence of UML ControlFlow element without a name).”

Sorry, I do not see this. The diagrams on the left in Figure 1 have three UML elements: a, c and d. Each occurs only once. Furthermore, “3 types of elements: UML Activity, UML UseCase, UML ControlFlow“, I do not see, for example, the control flow element.

What is the difference between ?Ì‚ and f, the f with the mark and the f without it? Line 156 states: “?? is the number of occurrences of elements of the i-th of the type i156= 0,1,2 ...” -> the i-th of the type i ? I do not understand.

Sorry, I do have a really hard time to understand what is done here. I think Section 3 needs to be reworked in its entirety, otherwise I cannot understand it. Hence, I am unable to rate the “scientific soundness” of the paper.

Round 2

Reviewer 1 Report

Although the authors have addressed some of the comments, it seems to me that the contribution is really minimal for a journal. I recommend submitting the work to a conference or reviewing the comments from the first review and extending the work.

Reviewer 2 Report

This is my second review of the paper. First of all, I want to thank the author for his detailed response to my comments, and also for updating the article. However, I do not feel that the issue has become much clearer. Instead, there are now more things that are not clear than before. See below. At its foundation, the article has some nice idea that is probably worth publishing (given that sometimes even seemingly trivial findings need publishing) but the article needs to make it much clearer and be less abstract. It appears that there are too many ideas and terms mixed in, obscuring what is actually important. While I like the conciseness of the paper, it would really benefit from explaining better for a reader’s “untrained eye”. I will hence recommend another major revision of this paper and really urge the author to try to make his many abstract concepts and ideas in this paper better understandable. Simplify. Please rigorously check the structure.

You might consider structuring your paper according to IMRaD. I expect that the paper would benefit very much. In particular, the research method should be sketched clearly (i.e., provide an overview of the approach how you do your proof) before going into too much details.

“Hence, a UML diagram with only one class does not contain any information about the given system. Only introducing another class and linking these classes with a relation, or describing this one class with attributes would bring some information to the IT architect. On the other hand, to encode a random class arrangement, every bit has to be encoded. Thus, an infinite stream would have infinite entropy. However, imagine that this class layout has no (useful) information. In this case, all the different random streams can be considered equivalent. There is nothing to encode.”

I do not agree that one class does not contain any information. It does not contain much information, right. But not “no information”. Two classes and one link does not add so much more information about the system.

Please explain in the paper: What you mean by random class arrangement and what bit has to be encoded. Next sentence, what stream do you refer to. Was it introduced before? Of course, an infinite stream will have infinite entropy, even if it is not random. What do you mean? I do not understand why there is a difference between useful and not useful information. I think entropy does not distinguish between useful and not useful. Why is that relevant to your proof? Why do equivalent streams have nothing to encode?

You introduce consistency rules as: “When constructing software architecture, IT architects more or less consciously however introduce elements that represent the same object instance on different diagrams with similar names. These connections are called consistency rules and are usually not saved in any way in the modeling tool”, i.e., connections specific to one system. Later, you mention that consistency rules are more generic constructs that can be published: “The first set of consistency rules (50 rules) was proposed in 2000 by Egyed [12] … ” This does not match.

In line 199 and line 204, there are direct quotes without reference. I suppose you are referring to what you wrote earlier.

It is not clear why AICC and CDE are used in the proof; and why edep and eindep need to use different formulas. Please introduce the basic idea of your proof (see IMRaD from above).